# Effect of communicating community immunity on COVID-19 vaccine-hesitant people from ethnically diverse backgrounds: an experimental vignette study in the UK

Sandro T Stoffel,[1,2] Aradhna Kaushal ![ORCID],[1] Aikaterini Grimani ![ORCID],[3] Christian von Wagner ![ORCID],[1] Falko F Sniehotta,[4,5] Ivo Vlaev ![ORCID] [3]

For numbered affiliations see end of article.

**Correspondence to**
Dr Aikaterini Grimani;
aikaterini.grimani@wbs.ac.uk

## ABSTRACT

**Objectives** Achieving high vaccination coverage is vital to the efforts of curbing the impact of the COVID-19 pandemic on public health and society. This study tested whether communicating the social benefit through community protection for friends and family members versus overall society, affects vaccination intention and perception among a sample enriched with respondents from black and ethnic minority backgrounds.

**Design** A web-based experimental survey was conducted. Eligible participants were individually randomised, with equal probability, to one of the three experimental vignettes.

**Setting** England.

**Participants** We recruited 512 (212 white, 300 ethnically diverse) vaccine-hesitant members from an online panel.

**Primary and secondary outcome measures** The primary outcome was the intention to get vaccinated against COVID-19. The secondary outcome consisted of a behavioural measure in the form of active interest in reading more about the COVID-19 vaccine. Additional measures included the perceived importance and expected uptake in others, as well as the attitudes towards vaccination.

**Results** Logistic regression models did not show an effect of the messages on intentions for the overall sample (*society*: adjusted OR (aOR): 128, 95% CI 0.88 to 1.88 and *friends and family*: aOR 1.32, 95% CI 0.89 to 1.94). The role of vaccination in achieving community immunity yielded higher vaccination intentions among study participants with white ethnic background (*society*: aOR: 1.94, 95% CI 1.07 to 3.51 and *friends and family*: aOR 2.07, 95% CI 1.08 to 3.96), but not among respondents from ethnically diverse backgrounds (*society*: aOR: 0.95, 95% CI 0.58 to 1.58 and *friends and family*: aOR 1.06, 95% CI 0.64 to 1.73). The messages, however, did not affect the perceived importance of the vaccine, expected vaccination uptake and active interest in reading more about the vaccine.

**Conclusions** Thus, although highlighting the social benefits of COVID-19 vaccinations can increase intentions among vaccine non-intenders, they are unlikely to address barriers among ethnically diverse communities.

## STRENGTHS AND LIMITATIONS OF THIS STUDY

⇒ This is one of the first experimental studies that investigated how the framing of the beneficiaries of community immunity influences vaccination intentions.
⇒ The study purposely recruited as many study participants from ethnic minorities as possible.
⇒ The study had a small analytical sample due to the difficulties in recruiting vaccine-hesitant study participants in the ongoing vaccination programme, limiting statistical power.
⇒ We had to combine all ethnic minorities so that nothing can be said about the impact of the messages on different ethnic minorities.
⇒ The study was conducted in May 2021, when the health policies during the pandemic were rapidly changing, limiting so the generalisability of the results.

## INTRODUCTION

The SARS-CoV-2 (COVID-19) pandemic has led to over 85 000 excess deaths in England and Wales and almost 6 million reported COVID-19 deaths between 1 January 2020 and 31 December 2021.[1 2] The success of the COVID-19 vaccination programme depends on high levels of coverage, around 80%, to achieve community immunity.[3] This is increasingly important as vaccine effectiveness wanes over time and may not necessarily confer protection against the new COVID-19 variants, requiring booster vaccinations to maintain a high level of coverage.[4] However, some subgroups of the population are more likely to be vaccine hesitant with uptake and intention varying by gender, age, education, employment status, deprivation and ethnicity.[5–7]

Data from the Office for National Statistics show higher rates of vaccine hesitancy in



younger age groups, women, people working in hospitality, personal services or transport sectors, people living in more deprived areas, and people from ethnic minority groups, particularly black/black British and Pakistani/Bangladeshi ethnic groups.[6 8 9] This is particularly concerning as many of these groups are at increased risk of contracting COVID-19, hospitalisation and death from COVID-19.[10–12] Reasons for vaccine hesitancy include worry about the side effects, worry about the long-term effects on health, waiting to see how effective vaccines are, and concerns about safety.[13]

A key method to promote vaccine uptake among individuals is to design public health messages to promote collective goals and use targeted messages as a mechanism to achieve the same wider collective aim, for example, save the National Health Service (NHS), protect lives, promote awareness of symptoms and control measures.[14–18] These 'protect each other' messages highlight the benefits of protective behaviours for the group and its most vulnerable members by promoting care for others rather than individual self-interest.[19] In addition, enlisting trusted voices has been shown to make public health messages more effective in changing behaviour during epidemics.[20 21]

While experimental studies on social preferences have shown that individuals are motivated by the well-being of others, individuals may hold social identities at various levels of abstraction, ranging from concrete groups of individuals (eg, own friends and family) to broader categories of individuals such as citizens of their country.[22 23] Studies on social preferences have shown that individuals exhibit more prosocial behaviour if they know more about the potential beneficiaries, such as their social belonging.[23–26] Thus, prosocial behaviour is negatively related to social distance; for example, whether beneficiaries are close friends or not.[27–29] As such, individuals' intentions to get immunised for the benefit of others may depend on the social distance to the beneficiaries of community immunity. A recent experimental study found that people in England had higher intentions to get vaccinated against a hypothetical influenza if they were told about the social benefit of vaccination for their close social environment, such as friends and family. Communicating the social benefit for the overall society, however, did not increase vaccination intentions.[30] Furthermore, as the cultural background can influence the perception of social benefits, communicating the concept of herd immunity can increase vaccination intentions among cultures that focus on collective benefits. Prosocial nudges can thus help to close these immunity gaps.[22]

The primary aim of the current research was to explore how the framing of community immunity influences vaccination intentions among white and ethnic minorities. For this, we conducted a web-based experiment and recruited vaccine-hesitant individuals with a white ethnicity and an ethnic minority background. Specifically, we investigated whether the definition of the beneficiaries of community immunity affects vaccination intentions, expecting that

highlighting the indirect effect of immunisation for the close social environment would increase intentions to get a COVID-19 vaccine. In line with previous research, we expect that the information about community immunity to be more effective in increasing vaccination intentions, if it mentions close social environments instead of overall society, as the definition of the beneficiaries influences social preferences.[30 31] Additionally, the study also investigated whether communicating community immunity influences the perception of the vaccine in terms of social importance and expected uptake of peers.

## METHODS

### General procedure

A web-based experimental survey was programmed in qualtrics to assess how the framing of community immunity influences vaccination intentions. The survey was conducted in May 2021 and featured a sample of adult men and women who were invited by a survey vendor (Dynata) to take part in an online survey on COVID-19 vaccination. Only study participants who completed the survey received a small financial incentive from the survey vendor, which was defined by the length of the questionnaire.

### Survey design

At the start of the survey, study participants were asked to give explicit consent for their data to be used and published as part of this research project before they could continue in the survey. A first screener question on screening behaviour was used to recruit individuals who had either been invited for the vaccination but declined it or who have not been invited yet but did not intend to get vaccinated when invited. The question was adapted from the NIHR Policy Research Unit Behavioural Sciences questionnaire on vaccination intentions. Participants were told that they would have the vaccine once it is available to them, with the response options; strongly disagree, disagree, neither agree nor disagree, agree and strongly agree.[32] Individuals who disagreed or neither agreed nor disagreed were then asked about their ethnicity. As vaccine hesitancy was identified as being common among ethnic minority groups, a quota was employed on white ethnicity to recruit no more than 215 respondents from a white ethnic background and at least 300 study participants with an ethnic minority background so that meaningful differences could be explored. Individuals who were filtered out from the survey were redirected to the study briefing and final survey page where they were thanked for their participation (see figure 1).

### Experimental conditions

Eligible individuals were individually randomised, with equal probability, to one of three experimental vignettes with different versions of the COVID-19 vaccination invitation letter. Each participant was told that the vaccination would be free and consisted of two separate shots

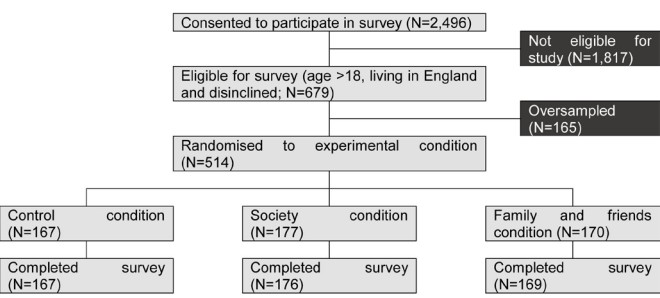

**Figure 1** Flow through experiment.

that would be administered in a vaccination centre. The type of vaccine was not mentioned in the vignettes. The last sentence of the vignette featured the experimental manipulation with the description of the beneficiaries of the vaccination (see table 1).

## Preregistration

The design and analysis plan for the experiment had been preregistered on Open Science Framework (OSF; https://osf.io/k4jyv/?view_only=08694ea40f994671b2dd899ae656f34e) before any data were collected. Questionnaire (see online supplemental file_questionnaire), analysis scripts and an abridged dataset containing all primary study variables can also be found on OSF.

## Outcome measures

### Vaccination intentions

The primary outcome was the intention to get vaccinated against COVID-19. This was measured using the question: 'Would you get vaccinated against the coronavirus?' with responses on a partially labelled 8-point scale, where 1 means that one will definitely not get vaccinated and 8 indicates that one will definitely get vaccinated.[30] The response options to these questions were taken from previous studies.[33 34] Individuals who stated low vaccination intentions (below 5) were asked to indicate reasons for not wanting to get vaccinated (see online supplemental table S1 for answers).

### Active interest

The secondary outcome consisted of a behavioural measure in the form of active interest in reading more about the COVID-19 vaccine.[35 36] Following the

**Table 1** Behavioural messages tested in the experiment

| Condition | Final paragraph of the vignette |
| --- | --- |
| Control | Get vaccinated to protect yourself from COVID-19 |
| Society | Get vaccinated to protect yourself and the vulnerable people in Great Britain from COVID-19 |
| Family and friends | Get vaccinated to protect yourself, your community, and your vulnerable friends and family members such as your children, your parents or your grandparents from the COVID-19 |

experimental manipulation towards the end of the survey, participants were asked whether they would like to 'read' or 'skip' information about the vaccination procedure before completing the survey. Those who indicated that they would like to skip it were sent to the end of the questionnaire with the sociodemographic question, while those who wanted to read the information were presented with additional information about the vaccination from the official website of the publicly funded national health service of England (NHS). Choosing to read the information was interpreted as an active interest in reading more health information. Engagement with the information was measured with four comprehension questions.

### Perceived importance and expected uptake in others

To explore the impact of the experimental manipulation on the perceived importance of the immunisation programme, we included the question. 'Do you believe that getting vaccinated against COVID-19 is socially important?'. Additionally, we asked whether they thought that most people would get immunised with the question 'Do you think that most people will get vaccinated?'. Both items featured 5-point fully labelled Likert scale response options.[30]

### Attitudes towards vaccination

Nine additional questions from the 5C measure of psychological antecedents of vaccination were included as further outcomes.[37] The questions featured 7-point fully labelled Likert scales (0–6) and were combined into three constructs on confidence in vaccination and public authorities, complacency and perceived collective responsibility with scores between 0 and 18 each.

### Sample size calculation and statistical analysis

The sample size for the experiment was based on estimates from a previous study.[30] A sample size calculation suggested minimum group sizes of 170 participants per group (510 participants in total).[30 38] We used a Kruskal-Wallis tests and multivariable ordered and binary logistic regressions adjusting for baseline vaccination invitation status and sociodemographic variables to investigate the effect of the experimental conditions on vaccination intentions and active interest. The regressions were run on the whole sample and the two ethnic subgroups; white and black, Asian and minority ethnic which describes non-white ethnicities. The statistical analysis was conducted with Stata/IC V.16.0 (StataCorp LP).

### Patient and public involvement

There was no patient or public involvement in the design, conduct and reporting of this research.

## RESULTS

### Study sample

The analytical sample comprised 512 individuals who had not yet been invited for a COVID-19 vaccination appointment but were planning not to accept the invitation

(N=302; 59.0%) or have been invited and had declined the vaccination (N=210, 41.0%). More than half of the study participants were from an ethnic minority (excluding white minorities) group (N=307, 59.6%), female (N=285, 55.7%) and between 25 and 34 years old (N=236, 46.1%). There were no statistically significant differences in socio-demographic characteristics, indicating that there were no imbalances due to levels of drop-out varying among the three experimental conditions (see online supplemental table S2 for the characteristics of the study sample).

### Vaccination intentions
The results of the ordered logistic regression, reported in table 2, showed that, compared with the control condition, neither the society nor the family and friends message yielded higher vaccination intentions (see online supplemental figure S1 for the distribution). Looking at white and ethnic minority subgroups separately reveals that the behavioural messages were associated with higher vaccination intentions in the former but not in the latter group (adjusted OR (aOR): 1.94; 95% CI 1.07 to 3.51 and aOR: 2.07; 95% CI 1.08 to 3.96).

### Active interest
The experimental manipulations did not influence interest in reading more about the vaccination process (see online supplemental table S3 for the regression results). Independently from the experimental condition, around one-third of the respondents stated that they wanted to read more (34.7–44.9%, $\chi^2$(2, N=512)=3.991, p=0.136). A Kruskal-Wallis test did not reveal any differences in vaccination knowledge across the conditions ($\chi^2$=0.655, p=0.721, df=2). Most participants (45.9%) who read the additional information answered two out of four comprehension questions correctly.

### Perceived importance and expected uptake in others
Only a minority of participants stated that they perceived the vaccinations as not important (N=129, 25.2%) and most thought the vaccination to be moderately important (N=163, 31.8%). Across the three conditions, the majority (N=271, 52.9%) expected that more than 60% of their peers would get vaccinated. Importantly, the experimental manipulations did not affect the perceived importance of the vaccination nor the expected uptake (see online supplemental tables S4, S5 and S6 for the regression results and online supplemental figures S2 and S3 for the distributions).

### Attitude towards vaccination
The median values for confidence in vaccination and public authorities was only 9 out of 18 (SD 4.84) and did not vary between experimental conditions ($\chi^2$=0.05, p=0.974, df=2). Similarly, complacency, the feeling that vaccination is not necessary was also low relatively with a median of 8 out of 18 (SD 4.51) and also no variation between experimental conditions ($\chi^2$=2.48, p=0.290, df=2). The median for perceived social responsibility was 11 out of 18 (SD 2.30) and was also not influenced by the

experimental manipulations ($\chi^2$=0.78, p=0.679, df=2). See online supplemental figure S4 for the distribution of the attitude scores.

### Reasons for vaccination hesitancy
Among the 306 study participants who did not intend to be vaccinated, fear of side effects and doubts about the protective nature of the vaccine were the most frequently mentioned reasons to not get vaccinated (66%) followed by not believing that the vaccine offers protection (38%). See online supplemental table S1 for a full list of the stated reasons.

## GENERAL DISCUSSION
This is one of the first studies to investigate how the framing of the beneficiaries of community immunity influences vaccination intentions.[30 31] In a web-based experimental study, we tested whether mentioning the social benefit of vaccination through community protection for friends and family members or overall society affects COVID-19 vaccination behaviour among white and ethnic minorities. Contrary to a previous study, Stoffel and Herrmann[30] adding this information only affected vaccination intentions among respondents from a white ethnic background. This lack of effect among respondents from ethnic minorities may be due to previously reported high levels of medical mistrust in ethnic minority groups which was found to mediate participation of black participants in COVID-19 vaccine trials and uptake more generally.[39] Future research could explore this further by testing messages relating to vaccine safety and efficacy.

### Limitations of the experimental study
Our study has several limitations which call for follow-up research. First, we only had a small analytical sample due to the difficulties in recruiting vaccine-hesitant study participants in the ongoing vaccination programme, limiting statistical power. Second, the study featured study samples from an online panel, which may limit the generalisability of the results. Third, we had to combine all ethnic minorities so that nothing can be said about the impact of the messages on different ethnic minorities. Fourth, we assessed vaccination intentions and willingness to read more about the vaccine. Therefore, the utility of communicating community immunity in changing vaccination behaviour cannot be determined as intentions do not necessarily translate to behaviours.[40 41] Thus, additional strategies may be required to build on motivational changes to increase vaccination uptake, such as implementation intentions.[42] Moreover, the study was conducted in May 2021, the health policies during the pandemic were rapidly changing, limiting so the generalisability of the results. A final limitation is that the survey did not include questions about the perceived risk of the responder and their family. It is possible that the different effects of the messages on white and ethnic minorities could have been

**Table 2** Ordered logistic regression on vaccination intentions[18]

| | Model 1: overall sample | | | | | Model 2: only white ethnic background | | | | | Model 3: black, Asian, mixed or other ethnic backgrounds | | | | |
|---|---|---|---|---|---|---|---|---|---|---|---|---|---|---|---|
| | Mean | OR | 95% CI | aOR | 95% CI | Mean | OR | 95% CI | aOR | 95% CI | Mean | OR | 95% CI | aOR | 95% CI |
| **Condition** | | | | | | | | | | | | | | | |
| Control | 3.60 | Ref. | | Ref. | | 3.42 | Ref. | | Ref. | | 3.74 | Ref. | | Ref. | |
| Society | 3.98 | 1.362 | 0.936 to 1.981 | 1.284 | 0.876 to 1.880 | 4.27 | 1.965 | 1.103 to 3.498* | 1.937 | 1.069 to 3.512* | 3.74 | 0.999 | 0.609 to 1.638 | 0.952 | 0.577 to 1.571 |
| Family and friends | 3.86 | 1.200 | 0.826 to 1.744 | 1.317 | 0.893 to 1.942 | 3.89 | 1.430 | 0.779 to 2.627 | 2.066 | 1.077 to 3.962* | 3.84 | 1.030 | 0.638 to 1.663 | 1.056 | 0.644 to 1.732 |
| **Vaccination status** | | | | | | | | | | | | | | | |
| Invited | 3.70 | Ref. | | Ref. | | 3.85 | Ref. | | Ref. | | 3.59 | Ref. | | Ref. | |
| Not invited | 3.98 | 1.263 | 0.929 to 1.718 | 1.192 | 0.859 to 1.654 | 3.90 | 1.035 | 0.636 to 1.683 | 0.678 | 0.394 to 1.169 | 4.02 | 1.450 | 0.974 to 2.161 | 1.516 | 0.993 to 2.315 |
| **Age (years)** | | | | | | | | | | | | | | | |
| 18–24 | 3.97 | Ref. | | Ref. | | 4.42 | Ref. | | Ref. | | 3.75 | Ref. | | Ref. | |
| 25–34 | 4.04 | 1.075 | 0.753 to 1.536 | 0.932 | 0.632 to 1.375 | 4.15 | 0.839 | 0.465 to 1.512 | 0.694 | 0.370 to 1.304 | 3.94 | 1.179 | 0.743 to 1.872 | 0.968 | 0.584 to 1.604 |
| 35–44 | 3.84 | 0.910 | 0.549 to 1.509 | 0.743 | 0.426 to 1.296 | 2.93 | 0.346 | 0.120 to 0.996* | 0.289 | 0.089 to 0.945* | 4.08 | 1.311 | 0.726 to 2.366 | 1.085 | 0.567 to 2.076 |
| 45+ | 2.58 | 0.297 | 0.173 to 0.508** | 0.289 | 0.161 to 0.519** | 2.24 | 0.182 | 0.077 to 0.430** | 0.163 | 0.064 to 0.416** | 2.86 | 0.423 | 0.209 to 0.855* | 0.347 | 0.159 to 0.755** |
| **Gender** | | | | | | | | | | | | | | | |
| Male | 4.08 | Ref. | | Ref. | | 4.25 | Ref. | | Ref. | | 3.93 | Ref. | | Ref. | |
| Female | 3.62 | 0.697 | 0.511 to 0.953* | 0.653 | 0.473 to 0.901** | 3.50 | 0.562 | 0.344 to 0.917* | 0.528 | 0.319 to 0.876* | 3.68 | 0.807 | 0.536 to 1.215 | 0.768 | 0.503 to 1.174 |
| Non-binary | 3.00 | 0.480 | 0.120 to 1.927 | 0.358 | 0.082 to 1.553 | 3.00 | 0.422 | 0.063 to 2.802 | 0.394 | 0.045 to 3.447 | 3.00 | 0.512 | 0.069 to 3.820 | 0.401 | 0.050 to 3.185 |
| **Ethnicity** | | | | | | | | | | | | | | | |
| White | 3.87 | Ref. | | Ref. | | | | | | | | | | . | |
| Mixed | 3.46 | 0.741 | 0.461 to 1.192 | 0.687 | 0.419 to 1.126 | | | | | | | | | . | |
| Asian | 4.42 | 1.509 | 0.998 to 2.281 | 1.477 | 0.959 to 2.277 | | | | | | | | | | |
| Black | 3.43 | 0.713 | 0.479 to 1.061 | 0.673 | 0.445 to 1.018 | | | | | | | | | | |
| Arab or other | 3.77 | 0.904 | 0.324 to 2.525 | 1.017 | 0.338 to 3.059 | | | | | | | | | | |
| **A-levels** | | | | | | | | | | | | | | | |
| No | 3.33 | Ref. | | Ref. | | 3.13 | Ref. | | Ref. | | 3.52 | Ref. | | Ref. | |

Continued

**Table 2** Continued

| | Model 1: overall sample | | | | | Model 2: only white ethnic background | | | | | Model 3: black, Asian, mixed or other ethnic backgrounds | | | | |
|---|---|---|---|---|---|---|---|---|---|---|---|---|---|---|---|
| | Mean | OR | 95% CI | aOR | 95% CI | Mean | OR | 95% CI | aOR | 95% CI | Mean | OR | 95% CI | aOR | 95% CI |
| Yes | 3.94 | 1.664 | 1.132 to 2.444** | 1.609 | 1.064 to 2.431* | 4.12 | 2.033 | 1.167 to 3.542* | 2.095 | 1.143 to 3.838* | 3.83 | 1.323 | 0.769 to 2.275 | 1.285 | 0.724 to 2.283 |
| Living condition | | | | | | | | | | | | | | | |
| Alone | 3.72 | Ref. | | Ref. | | 4.00 | Ref. | | Ref. | | 3.57 | Ref. | | Ref. | |
| With somebody | 3.92 | 1.170 | 0.861 to 1.591 | 1.207 | 0.861 to 1.693 | 3.76 | 0.840 | 0.519 to 1.361 | 0.806 | 0.485 to 1.341 | 4.07 | 1.533 | 1.025 to 2.293* | 1.815 | 1.143 to 2.884* |
| Paid employment | | | | | | | | | | | | | | | |
| No | 3.52 | Ref. | | Ref. | | 3.43 | Ref. | | Ref. | | 3.57 | Ref. | | Ref. | |
| Yes | 3.94 | 1.366 | 0.984 to 1.895 | 1.174 | 0.828 to 1.664 | 4.03 | 1.521 | 0.896 to 2.580 | 1.229 | 0.700 to 2.157 | 3.86 | 1.273 | 0.837 to 1.935 | 1.064 | 0.678 to 1.671 |
| N | | 512 | | 512 | | | 207 | | 207 | | | 305 | | 305 | |

*p<0.05; **p<0.01.
aOR, adjusted OR.

caused by differences in perceived risk. A recent study from the USA showed that racial and ethnic minorities are more likely to perceive COVID-19 as a major threat to the population and their individual health.[43]

## CONCLUSION

This study tested how the framing of community immunity affects intentions to get the COVID-19 vaccine among vaccine-hesitant individuals. Focusing on ethnic minority groups, the study investigated whether behavioural messages appealing to get vaccinated to protect oneself, family members and friends or the overall society increased vaccination intentions and interest in reading more about the vaccine. As the behavioural messages failed to affect the vaccination behaviour of vaccine-hesitant participants, future studies should develop interventions that are specifically targeted at reducing fear about side effects and concerns about the protective nature of the COVID-19 vaccine to achieve the highest possible level of vaccination uptake.

**Author affiliations**
[1]Department of Behavioural Science and Health, University College London, London, UK
[2]Institute of Pharmaceutical Medicine (ECPM), University of Basel, Basel, Switzerland
[3]NIHR Policy Research Unit in Behavioural Science – Behavioural Science Group, Warwick Business School, University of Warwick, Coventry, UK
[4]NIHR Policy Research Unit in Behavioural Science – Population Health Sciences Institute, Faculty of Medical Sciences, Newcastle University, Newcastle upon Tyne, UK
[5]Department of Public Health, Social and Preventive Medicine, Heidelberg University, Heidelberg, Germany

**Contributors** Conceptualisation: STS, CvW, AK, AG, FFS, IV; data curation: AK, STS; formal analysis: AK, STS, IV; investigation: STS, AG, CvW, AK, FFS, IV; methodology: STS, AG, CvW, AK, FFS, IV; supervision: FFS, IV; writing – original draft: STS, AK; writing – review and editing: STS, CvW, AK, AG, FFS, IV; acting as guarantor: AG. All authors read and approved the final manuscript.

**Funding** This project is funded by the National Institute for Health Research (NIHR) (Policy Research Programme (Policy Research Unit in Behavioural Science PR-PRU1217-20501)).

**Disclaimer** The views expressed are those of the author(s) and not necessarily those of the NIHR or the Department of Health and Social Care.

**Competing interests** None declared.

**Patient and public involvement** Patients and/or the public were not involved in the design, or conduct, or reporting, or dissemination plans of this research.

**Patient consent for publication** Not applicable.

**Ethics approval** This study involves human participants and was approved by University of Warwick's Humanities and Social Sciences Research Ethics Committee (HSSREC 142/20-21). Participants gave informed consent to participate in the study before taking part.

**Provenance and peer review** Not commissioned; externally peer reviewed.

**Data availability statement** Data are available upon reasonable request. The data presented in this study are available upon reasonable request from the corresponding author.

responsibility arising from any reliance placed on the content. Where the content includes any translated material, BMJ does not warrant the accuracy and reliability of the translations (including but not limited to local regulations, clinical guidelines, terminology, drug names and drug dosages), and is not responsible for any error and/or omissions arising from translation and adaptation or otherwise.

**ORCID iDs**
Aradhna Kaushal http://orcid.org/0000-0002-3815-0624
Aikaterini Grimani http://orcid.org/0000-0002-2076-6199
Christian von Wagner http://orcid.org/0000-0002-7971-0691
Ivo Vlaev http://orcid.org/0000-0002-3218-0144

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
