## [Reviewer comments · BMJ Open]

ARTICLE DETAILS

TITLE (PROVISIONAL)	The effect of communicating community immunity on COVID-19 vaccine-hesitant people from ethnically diverse backgrounds: an experimental vignette study in the UK
AUTHORS	Stoffel, Sandro T.; Kaushal, Aradhna; Grimani, Aikaterini; von Wagner, Christian; Sniehotta, Falko F.; Vlaev, Ivo

VERSION 1 – REVIEW

REVIEWER	Denford, Sarah University of Exeter, Sport and Health Science
REVIEW RETURNED	08-Jul-2022

GENERAL COMMENTS	Many thanks for the opportunity to review this work - I very much enjoyed reading the manuscript. The authors describe a web-based experimental survey to explore the impact of communicating community immunity on vaccine intention. The manuscript is exceptionally well written, very clear and precise, and a pleasure to read. The methods are comprehensive and the analysis is described clearly. The results are interesting. It is a shame that the study was not powered to allow for additional analyses – in particular, it would be interesting to explore any possible relationships between participants' perceptions of the vulnerability of themselves/their friends and families and the impact of messages. Overall a nice study and a good contribution to the literature.
---

REVIEWER	Sun, Caijun Sun Yat-Sen University
REVIEW RETURNED	14-Aug-2022

GENERAL COMMENTS	To improve the vaccination coverage is very important to control the COVID-19 pandemic on public health, and thus investigation of vaccination-related behaviors about vaccine hesitancy. This manuscript mainly investigated whether communicating community immunity on COVID-19 vaccine hesitant people will affect the following COVID-19 vaccination intention. Basically, this is an important topic needed to be addressed. However, there are numerous major and minor concerns regarding this manuscript. 1. One conclusion of this manuscript is that the information of communicating community immunity only affected vaccination intentions among respondents from a White Ethnic but not Black Ethnic participants. I am concerned this ethnic-related conclusion, especially about White Ethnic and Black Ethnic. Lack of the ethical approval information. 2. The study was conducted at May 2021, but the pandemic situation and vaccination policies changed greatly during 2022
--

	year. This change always influenced people's attitudes towards COVID-19 vaccination, and therefore the conclusion might be changed. 3. Only 512 participants (212 white, 300 ethnically diverse) were investigated. Why there are more ethnic minority people than White people in this study? Please state the inclusion and exclusion criteria for these participants in this survey. Selection bias may exist if the participants with small sample size were recruited without reasonable inclusion and exclusion criteria 4. Statistical methods can be mentioned in detail in the corresponding analysis Text. 5. Page 5, line 5-7: This information "over 18 million deaths worldwide" is wrong. Other data mentioned in the introduction need to be updated. 6. Page 5, line 52: define what is NHS? 7. Table 1 should be listed in three-line table. 8. The quality of English writing needs improvement.
--	---

VERSION 1 – AUTHOR RESPONSE

Reviewer 1

Many thanks for the opportunity to review this work - I very much enjoyed reading the manuscript. The authors describe a web-based experimental survey to explore the impact of communicating community immunity on vaccine intention. The manuscript is exceptionally well written, very clear and precise, and a pleasure to read. The methods are comprehensive and the analysis is described clearly. The results are interesting. It is a shame that the study was not powered to allow for additional analyses – in particular, it would be interesting to explore any possible relationships between participants' perceptions of the vulnerability of themselves/their friends and families and the impact of messages. Overall a nice study and a good contribution to the literature.

Author's response (AR): We thank the reviewer for the positive comments and the suggestion to look at the relationship between perceived vulnerability and impact of the messages. Unfortunately, our experiment did not feature any questions on the perceived vulnerability of the responders and their family.

We added an additional sentence to the limitation section that the questionnaire did not assess the perceived vulnerability of the responders and their family.

(p.15, line 11: "A final limitation is that the survey did not include questions about the perceived risk of the responder and their family. It is possible that the different effect of the messages on White and ethnic minorities could have been caused by differences in perceived risk. A recent study from the US showed that racial and ethnic minorities are more likely to perceive coronavirus as a major threat to population and their individual health [43].")

Reviewer 2

1. One conclusion of this manuscript is that the information of communicating community immunity only affected vaccination intentions among respondents from a White Ethnic but not Black Ethnic participants. I am concerned this ethnic-related conclusion, especially about White Ethnic and Black Ethnic.

AR: We thank the reviewer for this comment. Our study found that the messages about community immunity did increase vaccination intentions among study participants with a White ethnic background, but not with study participants with black, asian, mixed or other ethnic background. We state the discussion that the lack of effect among respondents from minority ethnic backgrounds may be due to previously reported high levels of medical mistrust in ethnic minority

groups. Additionally, while we did not assess perceived risk of the responder, a recent US study found that ethnic minorities are more likely to perceive COVID-19 as a major threat to population and their individual health.

2. Lack of the ethical approval information.

AR: Information about the ethical approval is included in the subsection "Ethics approval and preregistration" (p.8, line 1).

3. The study was conducted at May 2021, but the pandemic situation and vaccination policies changed greatly during 2022 year. This change always influenced people's attitudes towards COVID-19 vaccination, and therefore the conclusion might be changed.

AR: We thank the reviewer for this comment. We agree that the vaccination policies changed drastically in the last year. We state in the discussion section that the results refer to the 2021 vaccination policy and that attitudes towards vaccination may have changed since.

4. Only 512 participants (212 white, 300 ethnically diverse) were investigated. Why there are more ethnic minority people than White people in this study? Please state the inclusion and exclusion criteria for these participants in this survey. Selection bias may exist if the participants with small sample size were recruited without reasonable inclusion and exclusion criteria.

AR: We include further information about the recruitment process and the use of quotas to get as many study participants with ethical diverse background for the experiment. Specifically, a quota was employed on White ethnicity to recruit no more than 215 respondents from a White ethnic background and at least 300 study participants with an ethnic minority background so that meaningful differences could be explored.

Apart from vaccination hesitancy, no filter questions were included in the survey. A selection bias may have occurred as the study was conducted online and featured study participants from a panel. Panels are not representative of the population and mainly consist of high educated individuals. We included a sentence in the limitation section that the use of an online sample limits the generalisability of the study results.

(p.14, line 48: "Second, the study featured study samples from an online panel, which may limit the generalisability of the results")

5. Statistical methods can be mentioned in detail in the corresponding analysis Text.

AR: The statistical methods are mentioned in the sub-section on sample size calculation and statistical methods. There we state that we used A Kruskal-Wallis tests and multivariable ordered and binary logistic regressions.

6. Page 5, line 5-7: This information "over 18 million deaths worldwide" is wrong. Other data mentioned in the introduction need to be updated.

AR: We replaced the reference and cite now a study that the reported deaths were almost 6 million between 2020 and 2021.

7. Page 5, line 52: define what is NHS?

AR: We included a sentence that the NHS is the publicly funded national health service of England (NHS).

8. Table 1 should be listed in three-line table.

AR: We removed the horizontal dotted lines in the table.

9. The quality of English writing needs improvement.

AR: We proofread the article.